# Evaluating the In Vitro and In Vivo Prebiotic Effects of Different Xylo-Oligosaccharides Obtained from Bamboo Shoots by Hydrothermal Pretreatment Combined with Endo-Xylanase Hydrolysis

**DOI:** 10.3390/ijms241713422

**Published:** 2023-08-30

**Authors:** Junping Deng, Jinyan Yun, Yang Gu, Bowen Yan, Baishuang Yin, Caoxing Huang

**Affiliations:** 1Co-Innovation Center for Efficient Processing and Utilization of Forest Resources, College of Chemical Engineering, Nanjing Forestry University, Nanjing 210037, China; djp@njfu.edu.cn (J.D.); guyang513@njfu.edu.cn (Y.G.); yanbowen418@126.com (B.Y.); 2College of Animal Science and Technology, Jilin Agricultural Science and Technology University, Jilin 132109, China; yunjinyan2022@163.com

**Keywords:** bamboo shoots, Xylo-oligosaccharides, intestinal microflora, probiotics

## Abstract

Xylo-oligosaccharides (XOS) enriched with high fractions of X2-X3 are regarded as an effective prebiotic for regulating the intestinal microflora. In this study, the original XOS solution was obtained from bamboo shoots through hydrothermal pretreatment under optimized conditions. Subsequently, enzymatic hydrolysis with endo-xylanase was performed on the original XOS solution to enhance the abundance of the X2-X3 fractions. The results demonstrated that hydrothermal pretreatment yielded 21.24% of XOS in the hydrolysate solution, and subsequent enzymatic hydrolysis significantly increased the proportion of the X2-X3 fractions from 38.87% to 68.21%. Moreover, the XOS solutions with higher amounts of X2-X3 fractions exhibited superior performance in promoting the growth of probiotics such as *Bifidobacterium adolescentis* and *Lactobacillus acidophilus* in vitro, leading to increased production of short-chain fatty acids. In the in vivo colitis mouse model, XOS solutions with higher contents of X2-X3 fractions demonstrated enhanced efficacy against intestinal inflammation. Compared with the colitis mice (model group), the XOS solution with higher X2-X3 fractions (S1 group) could significantly increase the number of *Streptomyces* in the intestinal microflora, while the original XOS solution (S2 group) could significantly increase the number of *Bacteroides* in the intestinal microflora of colitis mice. In addition, the abundances of *Alcaligenes* and *Pasteurella* in the intestinal microflora of the S1 and S2 groups were much lower than in the model group. This effect was attributed to the ability of these XOS solutions to enhance species diversity, reversing the imbalance and disorder within the intestinal microflora. Overall, this work highlights the outstanding potential of XOS enriched with high contents of X2-X3 fractions as a regulator of the intestinal microbiota and as an anti-colitis agent.

## 1. Introduction

Xylo-oligosaccharides (XOS) are bioactive oligosaccharides formed through the β-1,4 glycosidic linkage of 2 to 10 xylose units. Xylobiose (X2), xylotriose (X3), xylotetracose (X4), xyloquinose (X5), and xylohexose (X6) constitute the primary constituents of XOS, among which X2 and X3 stand out as the principal bioactive components [1]. Currently, XOS is considered an important prebiotic that plays a crucial role in promoting the proliferation of beneficial bacteria in the intestinal tract, particularly *Bifidobacterium adolescentis* (*B. adolescentis*) and *Lactobacillus acidophilus* (*L. acidophilus*), which are essential for maintaining a healthy gut microecological balance in humans [2]. XOS exhibits substantial research potential relative to other prebiotics due to its cost-effectiveness, thermal and pH stability, sensory attributes, and multifaceted impacts on human health and livestock well-being [3]. Currently, XOS has been shown to inhibit the growth of harmful bacteria, such as *Escherichia coli*, *Clostridium perfringens*, and *Enterobacteriaceae*. Generally, colitis caused by harmful bacteria can result in inflammation and damage to the colon, leading to symptoms such as diarrhea, abdominal pain, and cramping. The presence of harmful bacteria in the gut microbiota can disrupt its balance and lead to changes in bowel habits. However, the administration of XOS as a feed has shown potential in restoring the balance of the gut microbiota and improving bowel habits affected by colitis. Therefore, XOS has been found to offer potential health benefits by improving gut health [4].

Generally, XOS can be obtained from various biomass sources, primarily from lignocellulosic biomass such as agricultural waste, wood, and bamboo [5]. The process of obtaining XOS involves hydrolyzing the hemicellulose component using specific enzymes (endo-xylanase or β-xylosidase) or chemical methods (mild acid hydrolysis and hydrothermal pretreatment). Among these lignocellulosic biomasses, bamboo shoots (BS) are the most promising substrate for preparing XOS due to their high content of carbohydrates and low amounts of lignin [6,7]. Traditionally, BS has been recognized as a key ingredient in domestic food preparations. The presence of abundant hemicelluloses in BS suggests its potential as a raw material for the production of XOS as a prebiotic, thereby enhancing its value and expanding its applications. However, limited research has been conducted to investigate methods of obtaining XOS from BS and to evaluate its prebiotic properties. Therefore, further studies are needed to explore effective methods for obtaining XOS from BS and assess its potential as a prebiotic.

To overcome the complex linkage between carbohydrates and lignin in the cell wall of bamboo species, various technologies can be employed to obtain xylan from BS for further degradation into XOS [8,9]. One approach involves enzymatic hydrolysis, a process in which the isolated xylan from BS is enzymatically broken down into XOS. Another approach involves the direct degradation of xylan from BS into XOS using acid treatment [10]. Acid pretreatment is widely used as a cost-effective and efficient method for producing XOS from various lignocellulosic biomasses. However, the use of acids can result in equipment corrosion and significant degradation of hemicellulose into xylose. Hydrothermal pretreatment is recognized as an environmentally friendly, cost-effective, and straightforward method for hemicellulose hydrolysis into XOS [11,12]. However, the resulting XOS obtained from biomass via hydrothermal pretreatment often contains a high proportion of high-degree XOS, which exhibits limited prebiotic activity. To overcome this issue, a combined approach involving hydrothermal pretreatment and enzymatic hydrolysis with endo-xylanase has been proposed to produce XOS with higher amounts of X2 and X fractions, which possess the best prebiotic properties for probiotics [13]. However, there is limited work exploring the application of XOS from BS as a prebiotic for modulating the gut microbiota in colitis, particularly in obtaining XOS preparations enriched with X2-X3 fractions.

In this work, XOS enriched with X2-X3 fractions (S1) was obtained through enzymatic hydrolysis of the original XOS solution (S2) derived from hydrothermally pretreated BS using endo-xylanase. In vitro experiments were conducted to assess the differential prebiotic effects of S1 and S2 on the proliferation of typical intestinal beneficial bacteria (*B. adolescentis* and *L. acidophilus*). Furthermore, the in vivo prebiotic effects of the different XOS solutions were evaluated by feeding enteritis-induced mice with S1 and S2 solutions. The histology of intestinal mucosal tissues and the growth of various flora in the intestines of the enteritis-induced mice were examined to assess the different effects of the XOS solutions.

## 2. Results and Discussion

### 2.1. Analysis of the Changes in Components of BS after Hydrothermal Pretreatment

It is well known that XOS can be acquired from xylan sourced from biomass through diverse approaches, such as mild acid hydrolysis or hydrothermal pretreatment [14,15]. In this study, the green, simple, and highly efficient technology of hydrothermal pretreatment was employed to obtain XOS from bamboo shoots (BS) at temperatures ranging from 150 °C to 190 °C. The aim was to determine the optimal pretreatment conditions that would yield the highest amount of XOS in the hydrolyzate (pretreatment solution). The changes in the major components of BS after pretreatment at different temperatures are analyzed and presented in Table 1.

Table 1 shows that the recovery yield of BS solid firstly decreased and then reached a plateau when the pretreatment temperature increased from 150 °C to 190 °C. This trend can be attributed to the varying degrees of degradation of glucan and xylan during hydrothermal pretreatment [16]. Particularly, the recovery yields of glucan decreased from 83.67% to 69.67%, whereas the removal efficiency of xylan increased from 51.00% to 86.49% as the pretreatment temperature increased. These findings align with the observations documented by Kim et al. [17]. In addition, it is seen that the removal yield of lignin decreased from 14.00% to 0.50% as the hydrothermal pretreatment temperature increased from 150 °C to 190 °C. This decline may be attributed to the formation of pseudo-lignin, a by-product of the condensation of degraded carbohydrates (glucose and xylose), which inevitably affects the final removal yield of total lignin from BS [18]. As the main focus of this work was to obtain XOS from BS by degrading xylan in the pretreatment hydrolyzate, further analysis will be conducted to determine the yield of XOS and XOS with X2-X6 fractions in the obtained hydrolyzate under different pretreatment conditions.

### 2.2. Increasing the X2-X3 Fractions in XOS from Hydrolyzate of BS after Hydrothermal Pretreatment by Enzymatic Hydrolysis

To evaluate the production of XOS in the hydrolyzate of BS following hydrothermal pretreatment, the yields of XOS were analyzed and are presented in Table 1. The results revealed an increase in XOS yield from 21.74% to 23.35% as the pretreatment temperature increased from 150 °C to 160 °C. However, further increasing the temperature from 160 °C to 190 °C led to a gradual decrease in XOS yield. The trend observed for the yield of the X2-X6 fraction in the XOS mirrored the overall XOS production, indicating that higher temperatures facilitate the degradation of xylan into XOS. Nevertheless, excessively high hydrothermal pretreatment temperatures can result in the direct degradation of xylan into monosaccharides or other degradation products, thereby reducing the yield of XOS. It is noteworthy that the X2-X6 fractions in XOS are recognized for their excellent prebiotic properties in regulating gut microbiota. Based on the yield of the X2-X6 fraction in the XOS obtained from the pretreatment hydrolyzate, it can be concluded that hydrothermal pretreatment at 160 °C demonstrated the best performance in producing the greatest amount of XOS.

It is well known that the XOS with higher amounts of X2-X3 possesses superior prebiotic properties in stimulating the growth of beneficial bacteria such as *Lactobacilli* and *Bifidobacteria* [19,20]. To further enhance the content of the X2-X3 fractions, enzymatic hydrolysis using endo-xylanase can be conducted on the hydrolyzate to degrade the xylan or XOS with high polymerization degree [21]. However, it is important to note that the presence of dissolved lignin fractions and phenolic compounds in the hydrolyzate can impede enzyme hydrolysis efficiency and impact the growth performance of beneficial bacteria [22]. Therefore, before enzymatic hydrolysis by endo-xylanase, the hydrolyzate at 160 °C was subjected to adsorption using XAD resin to remove these inhibitors. After purification, the distribution of X2-X3 in the hydrolyzate solution after enzymatic hydrolysis (S1) and the original hydrolyzate solution (S2) are presented in Table 2.

In Table 2, it can be seen that the purified S2 solution contained a total X2-X6 concentration of 1.33 g/L, with a relatively uniform distribution of X2-X6 fractions. However, through enzymatic hydrolysis using endo-xylanase, the total concentration of X2-X6 in the S1 solution was increased to 2.05 g/L. This increase can be attributed to the degradation of X4-X6 fractions and higher-degree XOS (>6) into X2-X3 fractions during the enzymatic hydrolysis process. As shown in Table 2, the proportion of X2-X3 in the S1 solution was 68.21%, which was significantly higher than that in the S2 solution (38.87%). Generally, XOS molecules with shorter chain lengths, such as X2-X3, are more readily utilized by probiotic bacteria in the gut, promoting their growth and metabolic activities. Hence, the enrichment of X2-X3 fractions in S1 through enzymatic hydrolysis provides valuable insights for the development of XOS-based interventions to improve gut health and promote the growth of beneficial gut bacteria.

### 2.3. Effects of Different XOS Solutions on the Proliferation of Bacteria and the Production of Organic Acids in the Culture System In Vitro

Prebiotics possess the capacity to augment advantageous gut bacteria without undergoing absorption or digestion by the gastrointestinal microbiota [23]. Once the population of probiotics in the gut attains a specific threshold, it can effectively regulate the abundance of intestinal microorganisms [24]. *B. adolescentis* and *L. acidophilus* are probiotics that not only foster intestinal well-being but also impede the proliferation of pathogens [10]. *B. adolescentis* plays a crucial role in breaking down complex carbohydrates and dietary fiber that cannot be digested by the host [25]. *L. acidophilus* has been shown to boost the immune system, reduce inflammation, and improve digestive health [26]. Both *B. adolescentis* and *L. acidophilus* have been extensively studied for their probiotic properties in relation to XOS derived from various biomass sources in vitro [5]. To evaluate the prebiotic ability of XOS solutions obtained from hydrothermal pretreatment of BS, S1 and S2 solutions, both containing X2-X6 at a concentration of 3 g/L, were employed as carbon sources to induce the growth of *B. adolescentis* and *L. acidophilus* in the culture medium. The effects of S1 and S2 on the growth of *B. adolescentis* and *L. acidophilus* are illustrated in Figure 1.

The results depicted in Figure 1a confirm that both *B. adolescentis* and *L. acidophilus* exhibited increased proliferation when cultured with XOS as the carbon source in the S1 and S2 solutions. After 72 h, the OD_600_ values for *B. adolescentis* cultured with S1 and S2 increased from 0.22 to 0.47 and 0.44, respectively. Similarly, when *L. acidophilus* was cultured with S1 and S2, the OD_600_ values increased from 0.23 to 0.46 and 0.30, respectively. These findings indicate that both S1 and S2 have a significant impact on the proliferation of *B. adolescentis* and *L. acidophilus* during anaerobic processes, which has also been reported in the works of Li et al. [27], Crittenden et al. [28], and Wang et al. [29]. Moreover, it was noted that XOS present in the S1 and S2 solutions exhibited superior efficacy in stimulating the growth of *B. adolescentis* compared with *L. acidophilus*. This disparity can be attributed to the heightened ability of *B. adolescentis* to secrete β-xylosidase, which can break down the β-(1,4) glycosidic bond in XOS to release the xylose for promoting the growth of intestinal bacteria [30]. Additionally, Figure 1a also reveals that the S1 solution exhibited better performance in promoting the proliferation of both *B. adolescentis* and *L. acidophilus* compared with the S2 solution, as evidenced by the increased OD_600_ values. This confirms that XOS derived from the original BS via hydrothermal pretreatment demonstrates favorable proliferation characteristics for probiotics. Notably, XOS containing higher amounts of X2-X3 fractions outperformed the original XOS solution obtained from the pretreatment hydrolyzate.

The metabolic activities of probiotics during their proliferation can be influenced by changes in pH in the surrounding environment. It is well recognized that short-chain fatty acids (SCFAs) are key metabolic products of intestinal bacteria stimulated by prebiotics. The increased production of SCFAs during the proliferation process can result in a decrease in micro-environmental pH, which is linked to bowel function, calcium absorption, and lipid metabolism [31]. Therefore, changes in pH values and the amounts of SCFAs (including lactic acid, acetic acid, propionic acid, and butyric acid) produced by *B. adolescentis* and *L. acidophilus* after being cultured with XOS solutions of S1 and S2 were measured and illustrated in Figure 1b and Figure 1c, respectively.

In Figure 1b, the pH value of the cultured medium for *B. adolescentis* displayed a rapid decrease within the initial 24 h, followed by a relatively stable range between 24 and 72 h. After 72 h of culture, the pH values of the medium decreased from 7.0 to 5.8 for the S1 solution and from 7.0 to 6.1 for the S2 solution. Similarly, for *L. acidophilus* cultured with S1 and S2 solutions, the pH values followed similar trends over the 0–72 h period, during which the pH values of the medium declined from 7.0 to 6.2 for the S1 solution and from 7.0 to 6.3 for the S2 solution. These results indicate that the medium pH values of *B. adolescentis* and *L. acidophilus* cultured with XOS in the S1 and S2 solutions decreased to varying extents, which may be attributed to their distinct abilities to produce SCFAs during the proliferation process [18,32]. Hence, further analysis will be conducted to explore the differences in production of SCFAs by *B. adolescentis* and *L. acidophilus*.

As depicted in Figure 1c, propionic acid and acetic acid were the primary SCFA produced by *B. adolescentis* cultured with S1 and S2 solutions after 72 h, with total concentrations of 3.00 g/L and 2.66 g/L, respectively. For *L. acidophilus* cultured with S1 and S2 solutions, propionic acid and acetic acid were also identified as the predominant acids among the produced SCFAs. The total concentrations of propionic acid and acetic acid were measured at 2.44 g/L and 1.52 g/L, respectively. These results indicate that *B. adolescentis* produced higher amounts of SCFAs compared with *L. acidophilus*, aligning with the observed decrease in pH values. Furthermore, it was observed that the S1 solution exhibited greater SCFA production for both *B. adolescentis* and *L. acidophilus* compared with the S2 solution. This difference can be attributed to the higher proportion of the X2-X3 fraction present in the XOS of the S1 solution compared with the S2 solution [33]. These findings align with the outcomes results documented by Kabel et al. [34].

Acetic acid is widely recognized for its ability to reduce the pH level in the intestinal tract, thereby promoting intestinal homeostasis. The presence of low pH and an acidic environment inhibits the growth of harmful bacteria and regulates the activity of enzymes within the intestinal system [35]. On the other hand, propionate plays a crucial role in regulating cholesterol production, which contributes to lowering the risk of cardiovascular disease. It also plays a significant role in supporting probiotic activity [36]. In this study, it was observed that both *B. adolescentis* and *L. acidophilus* produced SCFAs, primarily acetic acid and propionic acid. This finding suggests that the utilization of S1 and S2 solutions could have a significant prebiotic effect on the metabolism of these probiotics. It further indicates that XOS derived from the hydrothermal pretreatment of biomass possesses excellent prebiotic properties.

### 2.4. Effects of Different Solutions on the Morphological Symptoms of Colitis Mice Induced by E. coli In Vivo

It is widely recognized that XOS has demonstrated potential in treating colitis due to its positive impact on the gut microbiota and immune system. One of the key factors contributing to inflammation in colitis is an overactive immune response, and XOS has the ability to regulate immune cell function and cytokine production to reduce inflammation. To further investigate the in vivo probiotic capabilities of the XOS produced from BS, a colitis mouse model was induced by infection with *E. coli*. Then, the mice were fed with solutions of S1 and S2 at the same concentration on a daily basis, which were termed as S1 group and S2 group, respectively. The control groups included normal mice (blank group) and colitis mice without the administration of any XOS solution (model group). In this work, to visually assess the impact of the XOS solution on the recovery of the colons in the colitis mice, changes in colon length and histopathological symptoms of the colon tissues on 7th day were evaluated and are presented in Figure 2a and Figure 2b, respectively.

Figure 2a shows that the injection of *E. coli* led to a noticeable reduction in colon length among the colitis mice (model group), decreasing from 9.0 cm to 7.3 cm compared with the colon length of normal mice (blank group). However, the administration of XOS solutions of S1 and S2 effectively reversed the intestinal shortening in colitis mice and restored it to normal length. Furthermore, the colons of the control mice exhibited the formation of regular fecal particles. In contrast, loose fecal matter was observed in the colon of normal mice induced by *E. coli*. However, the state of the formed fecal particles in the colitis mice gradually recovered to the normal state, with the S1 solution demonstrating superior performance. This suggests that the XOS in the S1 solution, which contains a higher proportion of X2-X3 fractions, may exhibit better efficacy in restoring normal bowel movements and alleviating diarrhea in the colitis colon.

To evaluate the changes in colonic mucosal erosion and ulceration, the colonic tissue was fixed in paraffin and stained with hematoxylin-eosin. Figure 2b illustrates significant variations in the appearance and organization of the colon tissue of different groups. In the colitis mice group, the colonic mucosa was severely damaged, resulting in the loss of tissue structure and severe degradation of epithelial cells. Histopathological examination of the intestine revealed inflammation-related symptoms, including immune cell infiltration, epithelial cell destruction, and loss of crypts. Notably, when the colitis mice were treated with S1 and S2 solutions, these histological alterations were almost completely reversed. Furthermore, there were no observable signs of clinical toxicity in any of the mice exposed to S1 or S2 solutions. These findings suggest that the XOS in the S1 and S2 solutions can be effectively used as prebiotics to mitigate the damage to the intestinal tract in colitis mice. These results are consistent with the findings reported by Sheng et al. [37]. Overall, these studies provide evidence that the administration of XOS can improve colitis in mice by promoting gut health and reducing inflammation.

### 2.5. Effects of Different XOS Solutions on the Species Diversity and Differences in Intestinal Bacteria in Colitis Mice Induced by E. coli In Vivo

It is widely recognized that a symbiotic relationship exists between the intestinal microflora and the host, with a close association between the intestinal microflora and the pathogenesis of colitis [36]. In this study, the intestinal flora in colitis mice before and after administering the XOS solutions was analyzed using 16S rDNA amplifier sequencing. Then, a Venn diagram was created to visually represent the shared and unique operational taxonomic units (OTUs) among the four groups (Figure 3a). Compared with the model group (Model), each experimental group (S1 and S2) exhibited a unique set of 27 and 38 OTUs, respectively. Furthermore, compared with the blank group (Blank), each experimental group (S1 and S2) had 64 and 89 unique OTUs, respectively. The PCA diagram at the OTU level illustrates the changes in the flora structure (Figure 3b). When comparing the intestinal microflora between the model group (Model) and the experimental groups (S1 and S2), significant differences were observed. Notably, the disparity between the S1 group and the model group was more pronounced. Compared with the blank group (Blank), a significant difference was observed in the S1 group, whereas no significant difference was found in the S2 group. These results indicate significant differences in the intestinal microflora structure between the S1 and S2 groups and the model group, as well as between the S1 and blank groups. However, no significant difference was detected between the S2 and blank groups. Thus, it is suggested that administration of XOS solution enriched with X2-X3 fractions may exhibit improved efficacy against inflammation and restoring probiotics proliferation in the colitis mice induced by *E. coli*.

Intestinal microflora diversity encompasses a wide range of microorganisms residing in the gastrointestinal tract, including bacteria, viruses, fungi, and protozoa. These microorganisms play a crucial role in maintaining gut health and overall well-being [38]. α diversity refers to the diversity within specific regions or ecosystems [39]. The Chao1, Shannon, and Simpson alpha diversity indices are used to assess species diversity within a single sample [40]. To further investigate the impact of XOS in the S1 and S2 solutions on the intestinal microbiota of colitis mice, species diversity (α-diversity) was analyzed using the Chao index, Shannon index, and Simpson index, which evaluate the total species diversity, community diversity, and regional diversity, respectively, within the intestinal microbiota (Figure 3c). It can be observed in the figure that on the second day, both the Chao1 index and Shannon index of the S1 and S2 groups were higher compared with the model group (Model). Conversely, the Simpson index of the S1 and S2 groups was lower than that of the model group (Model). These results indicate that the administration of XOS in the solutions of the S1 and S2 groups contributed to an increased species richness within the intestinal microflora of the mice. Notably, S1 exhibited a greater efficacy in this regard compared with S2.

Furthermore, β diversity analysis was employed to examine the variation in diversity among different ecosystems. Comparing the analysis results in Figure 4d,e, notable distinctions in species composition and abundance at the OTU level were observed between the S1, S2 groups, and the model group (Model) on the 2nd and 4th day. Notably, the difference in abundance was more pronounced than the difference in species composition. Additionally, it was discovered that the microbiota in the intestinal microorganisms of colitis mice treated with different XOS solutions consisted predominantly of the phyla *Firmicutes*, *Bacteroides*, *Proteobacteria*, *Campilobacterota*, and *Actinobacteriota* [41]. As illustrated in Figure 3d, the dominant phyla in the blank group were *Firmicutes* (55.41%), *Bacteroides* (4.34%), *Proteobacteria* (32.15%), *Actinobacteria* (7.10%), and *Campilobacterota* (0.35%). At the genus level, significant changes in bacterial composition were observed in the model group. The relative abundance of *Proteobacteria* increased to 58.78%, while the relative abundance of *Chlamydia*, *Bacteroides*, *Actinobacteria*, and *Campilobacterota* decreased to 19.49%, 20.41%, 1.16%, and 0.06%, respectively. Furthermore, compared with the model group, the administration of XOS in the S1 and S2 solutions also influenced the bacterial composition at the genus level. In the S1 group, there was a significant increase in the relative abundance of *Streptomyces* to 91.90%, accompanied by a decrease in *Bacteroides* to 3.95% and *Proteobacteria* to 2.10%. On the other hand, in the S2 group, the relative abundance of *Streptomyces* decreased significantly to 25.26%, while the relative abundance of *Bacteroides* increased to 63.44%. Overall, this work revealed that the administration of both S1 and S2 solutions significantly increased the abundance of intestinal microflora in colitis mice, indicating that XOS can reverse the imbalance and disorder of intestinal flora. Moreover, XOS with higher amounts of the X2-X3 fraction exhibited better performance in this restoring ability.

Furthermore, Figure 3e depicts the relative abundance of specific genera in different groups. In the model group, the relative abundance of *Bacteroides*, *Blautia*, *Paenalcaligenes*, and *Alcaligenes* increased compared with the blank group. In the S1 group, the relative abundance of Lactobacillus showed an increase compared with the model group, followed by Streptococcus and *Rossiella*. On the other hand, in the S2 group, the relative abundance of *Bacteroides* increased compared with the model group, followed by *Lactobacillus* and *Muribaculaceae*. *Lactobacillus* is well-known for its ability to enhance intestinal protection against harmful bacteria, stimulate the production of human immune cells, and improve overall immunity. In comparison to the model group, the administration of the XOS solution in the S1 group significantly increased the abundance of *Lactobacillus*. This effect can be attributed to the higher content of X2-X3 fractions in the S1 solution, which has the ability to regulate the intestinal environment. Improved intestinal conditions allow *Lactobacillus* to thrive for longer periods. This finding is consistent with the research results of Han [42]. In addition, *Bacteroides* play a crucial role in regulating the intestinal immune system and maintaining the balance of microbial communities within the gut. This contributes to enhanced immunity and a reduced risk of infection [43,44]. The abundance of *Bacteroides* was significantly higher in the S2 group compared with the model group, indicating that the XOS solution in the S2 group was also effective in treating colitis in mice. Overall, the XOS in both the S1 and S2 solutions demonstrated significant control over the intestinal microflora and remedied the imbalance associated with colitis.

Furthermore, the species abundance heatmaps depicted in Figure 3f–h demonstrate that the abundance of *Alcaligenes* and *Paenalcaligenes* significantly decreased in the S1 and S2 groups compared with the model group. *Alcaligenes* and *Paenalcaligenes* are Gram-negative bacteria belonging to the *Gammaproteobacteria* class of the *Proteobacteria* phylum. These bacteria have the potential to cause infections and pose a risk, particularly for individuals with weakened immune systems. The findings suggest that the administration of XOS in the S1 and S2 solutions can enhance the immune response of mice with enteritis and promote the balance of intestinal microflora by reducing the abundance of *Alcaligenes* and *Paenalcaligenes* in colitis mice. These results demonstrate that XOS produced through hydrothermal pretreatment not only helps maintain the balance of intestinal microflora for optimal intestinal health but also inhibits the alterations in intestinal microflora associated with colitis. Notably, XOS enriched with the X2-X3 fraction exhibited the most favorable performance.

### 2.6. Functional Evaluation of the Effects of Different XOS Solutions on the Intestinal Microflora in Colitis Mice

The microflora community within the intestine plays various roles in maintaining overall health. It can influence the rate of digestion and absorption of disaccharides, the metabolism of fats, and the metabolism of glucose [45]. These factors can have implications for metabolic health and weight regulation in the body. To predict the function or phenotype of microflora communities, several approaches are utilized based on the results obtained from high-throughput sequencing of prokaryotic 16S rDNA. Four commonly used approaches are PICRUSt, Tax4Fun, FAPROTAX, and BugBase. These methods provide insights into the functional potential of the microflora community.

In this study, the PICRUSt approach was used to predict the functional potential of the mice’s intestinal microflora community. As depicted in Figure 4, the results demonstrated that, compared with the model group, the S1 and S2 groups exhibited significant up-regulation in sugar biosynthesis, sugar metabolism, and amino acid metabolism. The up-regulation of sugar biosynthesis and metabolism suggests that certain sugar molecules may function as cellular signal molecules, controlling the growth and metabolism of intestinal microflora during their synthesis and breakdown [46]. Amino acid metabolism is vital for the breakdown and utilization of various amino acids by the intestinal microflora, leading to the production of intermediate metabolites such as fatty acids, organic acids, and aromatic amino acids. These metabolites play a crucial role in maintaining the balance and homeostasis of the intestinal microflora [45]. Based on the findings, it can be inferred that the administration of XOS in the S1 and S2 solutions leads to a significant up-regulation of sugar biosynthesis and metabolism as well as amino acid metabolism. This suggests that the XOS solutions from BS have the potential to maintain the balance and homeostasis of the intestinal environment.

## 3. Materials and Methods

### 3.1. Materials

The used bamboo shoots (12.37% glucan, 8.94% xylan, and 21.73% lignin) in this experiment were obtained in the bamboo forest in Linan, China. *B. adolescentis* (CICC6070) and *L. acidophilus* (CICC6074) strains were purchased from the China microbial strain Preservation Management Committee. The industrial endo-xylanase (activity of 20 U/mL) derived from *Trichoderma reesei* were provided by Jiangsu Kangwei Biotechnology Co., Ltd. (Yanchen, China). All the chemical reagents used in this study were purchased from Nanjing Chemical Reagent Co., Ltd (Nanjing, China). and did not require additional purification.

### 3.2. Preparing the Different XOS Solutions from BS

To determine the optimal conditions for obtaining the XOS solution from BS, 7 g of dried BS was mixed with 70 mL of distilled water in a 100 mL reactor. The mixture was then subjected to pretreatment at temperatures of 150 °C, 160 °C, 170 °C, 180 °C, and 190 °C for 1 h. After pretreatment, the reactor was cooled to room temperature using cold water, and solid–liquid separation was performed. The collected pretreatment solutions from the different conditions were analyzed to determine the XOS content and identify the best condition for XOS production from BS.

To remove the dissolved lignin in the solutions, a purification step was performed using XAD-16N resin. The resin was packed into a glass column for passing of the XOS solutions through the resin. During this process, the lignin fractions were adsorbed onto the resin, while the XOS solution was eluted, resulting in a purified XOS solution.

To enrich the X2-X3 fractions, XOS solution (S1) was prepared through enzymatic hydrolysis of the original XOS solution (S2) obtained from the optimal pretreatment condition. The enzymatic hydrolysis was performed using endo-xylanase according to the method described by Su et al. [47]. Specifically, 60 mL of S2 solution was mixed with 9 mL of endo-β-1,4-xylanase in a 100 mL conical flask. The enzymatic hydrolysis was carried out at a pH of 4.8, at 50 °C, and the solution was mixed at 150 rpm for 12 h.

### 3.3. Cultivation of B. adolescentis and L. acidophilus with Different XOS Solutions

To evaluate the in vitro differential prebiotic effects of the prepared XOS solutions, *B. adolescentis* and *L. acidophilus* were cultured using the S1 and S2 solutions as carbon sources. The XOS solutions, with a total concentration of X2-X6 oligosaccharides of 3 g/L, were utilized for culturing at 37 °C for 36 h in an anaerobic environment. The activation and cultivation of *B. adolescentis* and *L. acidophilus* followed the methods described by Huang et al. [48]. The pH of the activated and cultivated media for *B. adolescentis* and *L. acidophilus* was maintained at 7.0 and 6.8, respectively. Samples were collected at 6, 12, 24, 48, and 72 h to assess the proliferation of bacteria, pH variations, and organic acid production in the media. All the experiments were performed in duplicate for rigorous analysis.

### 3.4. Analysis of Bacterial Growth, pH Value and Organic Acid Content of the Media

The optical density (OD) values of *B. adolescentis* and *L. acidophilus* in the fermentation media were evaluated at 600 nm to evaluate the growth gut bacteria resulting from the prebiotic XOS solutions [49]. The pH values of the fermentation media were checked on a regular basis using a pH meter (PE-20K, Mettler Toledo, Switzerland, Zurich). An HPLC (Agilent 1260, Santa Clara, CA, USA) system was used to measure the levels of organic acids (lactic acid, acetic acid, propionate, and butyric acid) in the fermentation media. The HPLC system was equipped with a column of Aminex Bio-RAD (Hercules, CA, USA) HPX-87H and a differential refraction detector. For the mobile phase, 5 mM H_2_SO_4_ was used with flow rate of 0.6 mL/min at 55 °C.

### 3.5. Establishment and Treatment of Experimental Colitis Mouse Model

The in vivo growth performance resulting from the prepared XOS solutions was evaluated using a colitis mouse model induced by injection of *E. coli* (0.2 × 109 cfu). The colitis mice were orally administered 0.2 mL of either the S1 or S2 solution on a daily basis, along with normal drinking water and regular food. The 2-week-old Balb/c male mice with a weight range of 25 ± 2 g were obtained from Jiutai Small Animal Supplies Trading Co., Ltd. (Changchun, China). The care and experimental protocols involving the mice were approved by the Animal Ethics Committee of Jilin Agricultural Science and Technology University, and the study was conducted in accordance with their ethical guidelines (ethical qualification number: AWEC2017A01).

For the experiments, a total of 24 male mice were randomly divided into four groups (*n* = 6/group): the normal group (normal mice fed with sterile water), colitis group (colitis mice fed with sterile water), S1 group (colitis mice fed with sterile water and S1 solution), and S2 group (colitis mice fed with sterile water and S2 solution). The effects of the XOS solutions on the intestinal microflora were investigated by analyzing the length and histopathology of the colon in six mice sacrificed on the 2nd, 4th, and 7th day. In addition, the intestinal microflora diversity was analyzed using 16s rDNA sequencing, which was performed by Shanghai Meiji Biological and Pharmaceutical Technology Co., Ltd. (Shanghai China). Specifically, the genomic DNA was extracted according to the instructions of the Wizard^®^ Genome DNA purification Kit (Promega, Madison, WI, USA). The purified genomic DNA was quantified by TBS-380 fluorimeter (Turner BioSystems Inc., Sunnyvale, CA, USA). The genome sequencing combined PacBio RS II single molecule real-time sequencing (SMRT, Huntington Park, CA, USA) and the Illumina sequencing platform. Following bridge PCR amplification, double-ended sequencing (2 × 150 bp) was performed on the Illumina NovaSeq/Hiseq Xten (Illumina, San Diego, CA, USA) sequencing platform using the standard protocols in the analysis system.

### 3.6. Analysis Methods

The difference in xylose concentration before and after acid hydrolysis was used to calculate the XOS content in the pretreatment solutions. Specifically, sulfuric acid was added to the XOS solutions to achieve a 4% acid content before reacting at 121 °C for 90 min. The quantities of xylose in the pretreatment solution, acid hydrolysis solution, and organic acid-containing fermentation media were determined using an HPLC system (Agilent 1260) coupled with an Aminex Bio-Rad HPX-87H column, as described by Huang et al. [48]. A 5 mM H_2_SO_4_ solution was used as the mobile phase at a flow rate of 0.6 mL/min. Prior to analysis, all the samples were diluted with Milli-Q water. Each experiment was conducted in duplicate for accuracy and reliability.

The concentrations of X2-X6 in the S1 and S2 solutions were determined using an HPAEC system (Dionex ICS 3000, Milpitas, CA, USA) equipped with a CarboPac PA200 column and a pulse amperometric detector. The analysis was performed under the following conditions: column temperature, 30 °C; a mobile phase consisting of 0.1 M NaOH (A) and 0.5 M NaOAc (B). The flow rate was 0.3 mL/min with a gradient elution progress as follows: 0–7 min~100% A, 7–45 min~A reduced the gradient from 100% to 65%, B increased from 0% to 35%, 45–50 min~65% A + 35% B, and 50–65 min~100% A. Two parallel assays were conducted for each experiment to ensure accuracy and reliability.

The 16s rDNA sequencing analysis results were analyzed using various bioinformatics tools. Alpha diversity analysis was performed using Mothur 1.30.2 to assess the diversity within individual samples. Beta diversity calculations were generated using Qiime 1.9.1 to compare the differences in microbial community composition between samples. OTU clustering was performed using Uparse 7.0.1090 to group similar sequences into operational taxonomic units (OTUs), and OTU statistics were generated using Usearch 7 to analyze the abundance and distribution of OTUs in the samples.

## 4. Conclusions

This study provides evidence that the X2-X3 fractions within the original XOS solution obtained through BS hydrothermal pretreatment can be enhanced by endo-xylanase hydrolysis. XOS with a higher content of the X2-X3 fraction demonstrated superior performance in promoting the proliferation of *B. adolescentis* and *L. acidophilus*, leading to increased production of SCFAs in vitro. Moreover, the administration of XOS enriched with the X2-X3 fraction exhibited improved efficacy against intestinal inflammation in colitis mice in vivo. This can be attributed to its ability to enhance species diversity, thereby addressing the imbalance and disorder within the intestinal microflora. The findings of this study suggest that utilizing XOS derived from BS as a potential prebiotic has the potential to serve as a novel approach to regulating the intestinal microbiota and inhibiting colitis in individuals with enteritis.

## Figures and Tables

**Figure 1 ijms-24-13422-f001:**
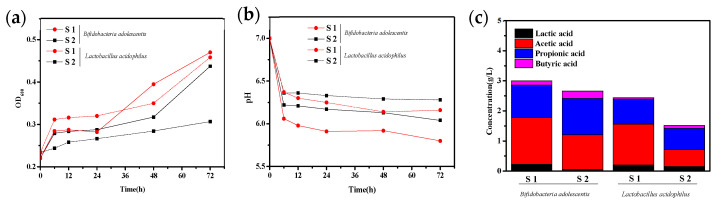
Effects of S1 (after enzymatic hydrolysis) and S2 (before enzymatic hydrolysis) on OD, pH and long acid content of *B. adolescentis* and *L. acidophilus*. OD_600_ (**a**), pH (**b**) and accumulation of organic acid metabolites of intestinal probiotics (**c**).

**Figure 2 ijms-24-13422-f002:**
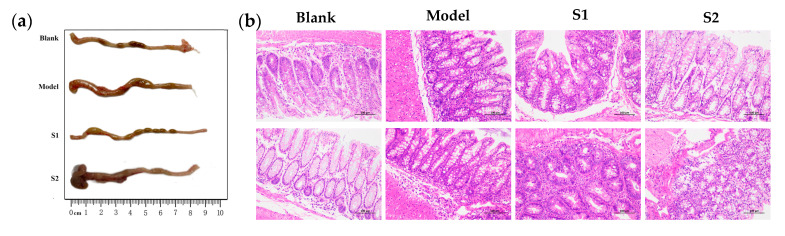
S1 and S2 ameliorated the morphological symptoms of acute colitis in DSS-induced mice. Colon length (**a**) and representative sections after H&E staining (**b**).

**Figure 3 ijms-24-13422-f003:**
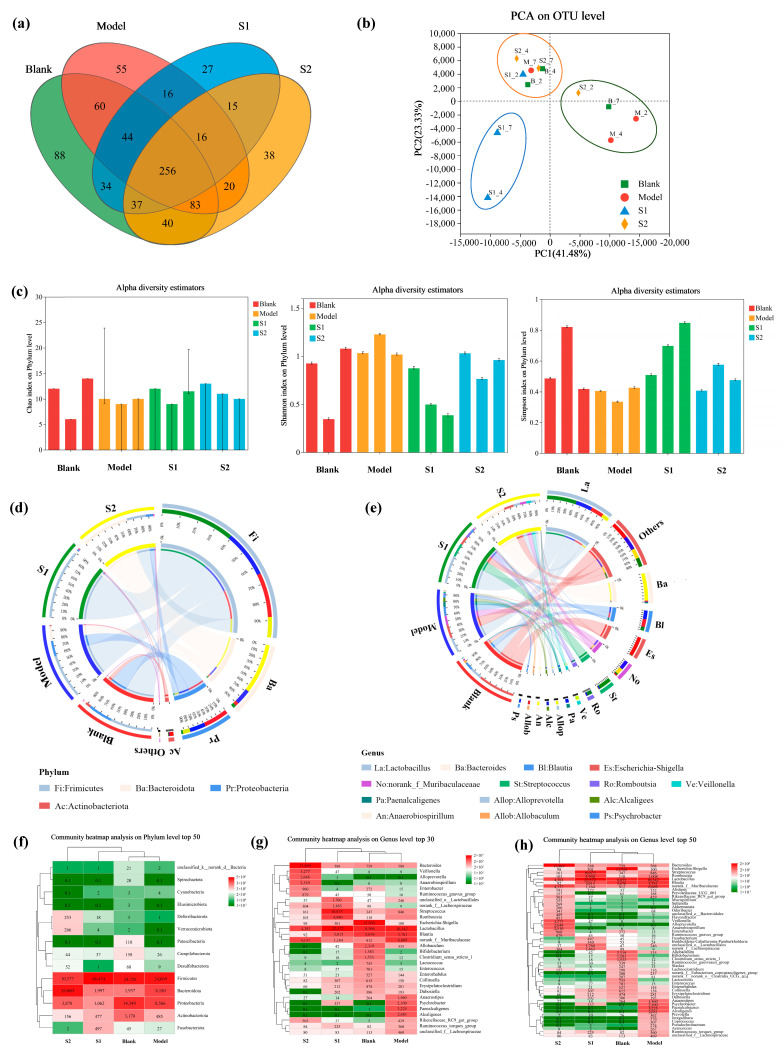
Venn diagram (**a**), principal component analysis (PCA) (**b**), and the alpha diversity index (**c**) of the intestinal microflora of different XOS treatment groups. Visualization circle diagram of the relationship between samples and species (**d**,**e**). Heatmaps of sample community composition on day 7 (**f**–**h**). Phylum level (**d**,**f**); genus level (**e**,**g**,**h**).

**Figure 4 ijms-24-13422-f004:**
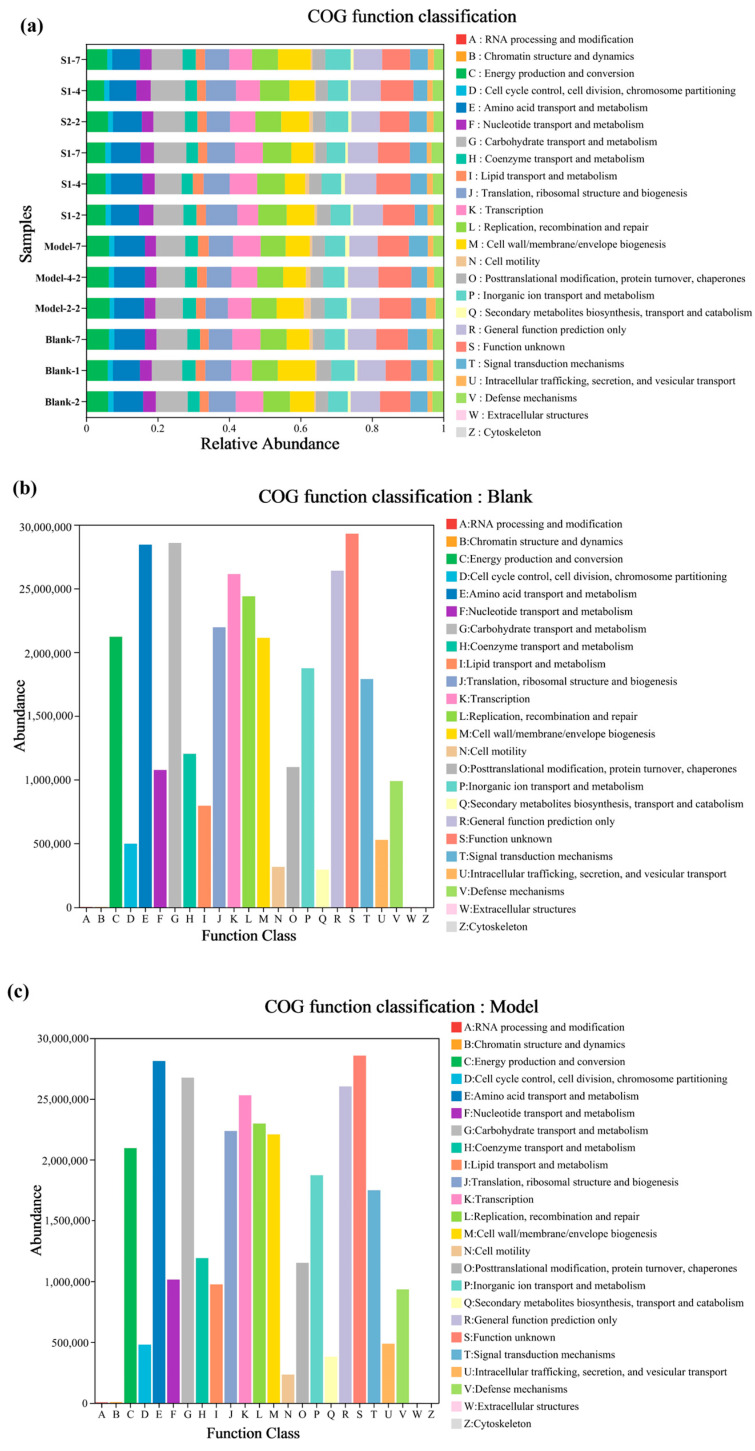
Relative abundance of PICRUSt inferred functions. Bar graph of COG functional classification statistics for all samples (**a**); bar graph of single-sample COG functional classification on day 7; (**b**) blank, (**c**) model, (**d**) S1 and (**e**) S2.

**Table 1 ijms-24-13422-t001:** Composition analysis of BS after hydrothermal pretreatment and the XOS yields at different temperatures (%).

Tem.	Glucan	Xylan	Lignin	Recovery Yield		Removal Yield		X2-X6 Yield	XOS Yield
				Solid	Glucan	Xylan	Lignin		
Raw material	12.37 ± 0.15	8.94 ± 0.09	21.73 ± 0.21	/	/	/	/	/	/
150 °C	23.20 ± 0.11	9.62 ± 0.00	41.89 ± 1.23	44.61	83.67%	51.00	14.00	19.10	21.74
160 °C	23.98 ± 0.41	6.62 ± 0.01	52.78 ± 0.81	38.72	75.06%	71.33	5.95	21.24	23.35
170 °C	24.79 ± 0.21	4.78 ± 0.00	58.52 ± 0.32	36.08	72.31%	80.71	2.83	18.50	19.47
180 °C	25.01 ± 0.10	4.31 ± 0.20	61.61 ± 0.34	34.89	70.54%	83.18	1.08	9.25	11.59
190 °C	24.90 ± 0.04	3.49 ± 0.11	62.47 ± 0.06	34.61	69.67	86.49	0.50	6.01	6.75

**Table 2 ijms-24-13422-t002:** The concentration of X2-X6 fractions in XOS solutions from hydrolyzate before and after enzymatic hydrolysis (S1) by endo-xylanase (g/L).

Solution	X2-X6 Concentration					
	X_2_	X_3_	X_4_	X_5_	X_6_	Total
S2	0.27 ± 0.08	0.25 ± 0.00	0.34 ± 0.04	0.19 ± 0.00	0.28 ± 0.11	1.33
S1	0.74 ± 0.10	0.67 ± 0.01	0.37 ± 0.01	0.10 ± 0.00	0.19 ± 0.04	2.07

## Data Availability

Not applicable.

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
