# Peer review of "Evaluating the In Vitro and In Vivo Prebiotic Effects of Different Xylo-Oligosaccharides Obtained from Bamboo Shoots by Hydrothermal Pretreatment Combined with Endo-Xylanase Hydrolysis"

_ijms, 2023, doi:10.3390/ijms241713422_

Round 1
Reviewer 1 Report
Dear Authors,
The manuscript of J. Deng et al, “Evaluating the in vitro and in vivo probiotics ability of different xylo-oligosaccharides from bamboo shoots by hydrothermal pretreatment combined with endo-xylanase hydrolysis” deals with preparation and in vitro and in vivo testing of xylooligosaccharide-based prebiotics from bamboo shoots. Hydrothermal pretreatment followed by enzymatic hydrolysis significantly increased the proportion of targeted X2-X3 XOS from 38.87% to 68.21%. XOS solutions with higher content of X2-X3 showed superior performance in promoting the growth of probiotics such as Bifidobacterium adolescentis and Lactobacillus acidophilus in vitro, leading to increased production of short-chain fatty acids. In vivo experiments with the colitis mouse model demonstrated that XOS solutions with higher content of X2-X3 XOS enhanced efficacy in intervening with intestinal inflammation. The authors explained this effect as ability of such XOS to enhance species diversity addressing the imbalance and disorder within the intestinal microflora. Thus, highly promising potential of X2-X3 XOS as regulators of the intestinal microbiota and anti-colitis agents was evaluated. The quality of manuscript looks OK though minor changes and corrections are needed.
Reviewer's notes. 1. Species names should be italicized; please, check this over the text.
2. Line 148, Table 2, title. "be Figure S2": what does it mean? ("before" or something else?) Please, correct.
3. Line 149, Line 2 in Table 2. Total should be 2.07.
4. Line 150. Title of a subsection. "Ef 2.3": something is erroneously inserted? Please, correct.
5. Lines 447—448. Evidently, pH values are measured by pH meter, not "conductivity meter". Please, specify what instrument was used.
6. Line 448. Extra capitalization (...using...).
7. Lines 448—450. "An HPLC (Agilent 1260) system was used to measure the level of organic acids...": please, specify conditions (column, solvent system, detection, etc. — and/or give the reference).
8. Lines 482—483. What is a concentration of NaOH in the mobile phase? Is this mobile phase isocratic or gradient? Please, clarify.
9. Lines 515 and below, References. Extra hyphenations were found in Refs. 17, 20, 23, 28, 32, 37, 40, 44, and 46. Please, correct.
10. Line 612, ref. 42. Journal title (Nature Reviews. Immunology) is missed.

Author Response
Dear Reviewers:
We would like to thank you for taking time to review our manuscript. We have revised the manuscript according to your comments. We have sent the revised manuscript and highlighted the changes by red. The main corrections in the paper and the responses to the your comments are as following:
- The manuscript of J. Deng et al, “Evaluating the in vitro and in vivo probiotics ability of different xylo-oligosaccharides from bamboo shoots by hydrothermal pretreatment combined with endo-xylanase hydrolysis” deals with preparation and in vitro and in vivo testing of xylo-oligosaccharide-based prebiotics from bamboo shoots. Hydrothermal pretreatment followed by enzymatic hydrolysis significantly increased the proportion of targeted X2-X3 XOS from 38.87% to 68.21%. XOS solutions with higher content of X2-X3 showed superior performance in promoting the growth of probiotics such as Bifidobacterium adolescentisand Lactobacillus acidophilusin vitro, leading to increased production of short-chain fatty acids. In vivo, experiments with the colitis mouse model demonstrated that XOS solutions with higher content of X2-X3 XOS enhanced efficacy in intervening with intestinal inflammation. The authors explained this effect as ability of such XOS to enhance species diversity addressing the imbalance and disorder within the intestinal microflora. Thus, highly promising potential of X2-X3 XOS as regulators of the intestinal microbiota and anti-colitis agents was evaluated. The quality of manuscript looks OK though minor changes and corrections are needed. Species names should be italicized; please, check this over the text.
Response:
Thanks for your comment. We have changed the name of the species to italics, see lines 171, 340, 385, etc. Please check the changes.
- Line 148, Table 2, title. "be Figure S2": what does it mean? ("before" or something else?) Please, correct.
Response:
Sorry for our mistakes. "be Figure S2" should be "before". We have corrected it, see lines 164. Please check the changes.
- Line 149, Line 2 in Table 2. Total should be 2.07.
Response:
Thanks for your comment. We have changed the total to 2.07, see Line 165, Line 2 in Table 2. Please check the changes.
- Line 150. Title of a subsection. "Ef 2.3": something is wrongly inserted? Please, correct.
Response:
Sorry for our mistakes. We have deleted "Ef 2.3", see lines 166. Please check the changes.
- Lines 447—448. Evidently, pH values are measured by pH meter, not "conductivity meter". Please, specify what instrument was used.
Response:
Sorry for our mistakes. We have changed "conductivity meter" to " pH meter", see lines 470. Please check the changes.
- Line 448. Extra capitalization (...using...).
Response:
Thanks for your comment. We have corrected the case of letters, see lines 470. Please check the changes.
- Lines 448—450. "An HPLC (Agilent 1260) system was used to measure the level of organic acids...": please, specify conditions (column, solvent system, detection, etc. — and/or give the reference).
Response:
Thanks for your comment. We have added a note to the statement in the manuscript, see lines 472-475. Please check the addition.
- Lines 482—483. What is a concentration of NaOH in the mobile phase? Is this mobile phase isocratic or gradient? Please, clarify.
Response:
Thanks for your comment. The analysis was performed under the conditions of column temperature of 30 °C, mobile phase consisting of 0.1 M NaOH (A) and 0.5 M NaOAc (B), flow rate of 0.3 mL/min at gradient elution progress (0-7 min~100%A, 7-45 min~ A reduced the gradient from 100% to 65%, B increased from 0% to 35%, 45-50 min~65%A+35%B, 50-65 min~100%A). We have added the condition of exoneration to the manuscript, see lines 515-517. Please check the addition.
- Lines 515 and below, References. Extra hyphenations were found in Refs. 17, 20, 23, 28, 32, 37, 40, 44, and 46. Please, correct.
Response:
Sorry for our mistakes. We have corrected it, see lines 599 and below. Please check the changes.
- Line 612, ref. 42. Journal title (Nature Reviews. Immunology) is missed.
Response:
Sorry for our mistakes. We have corrected it, see lines 656. Please check the changes.
Reviewer 2 Report
Manuscript is very related to the current topic - preparation prebiotics based on degradation products of plant polysaccharides and is fully consistent with IJMS. The paper is well written, organized and adds new understanding to the xylo-oligosaccharides production.
Abstract
Despite most of the research related to the impact of the resulting prebiotics on the microflora, the abstract provides an insufficient amount of first-time results.
Introduction
Given the relevance of the ongoing research, there are not enough references in the introduction. An explanation is required why the authors are focused specifically on xylooligosaccharides, and not, say, for example manno- derivatives? Why exactly X2-X3 needs to be explained in more detail. You can give an example of obtaining from raw materials as close as possible to bamboo.
Results and discussion
A discrepancy was found in the value of the component composition of raw materials Table 1 (first line) and Introduction (line 48).
Materials and methods
It is required to indicate in more detail what kind of enzyme preparation of xylanase was used, what is the activity? Industrial or laboratory? Were there any side activities?
Were other (non xylo-) oligosaccharides identified?
How was DNA isolated from microflora? What is the 16S rRNA gene fragment? On what equipment was sequencing carried out, according to what protocol?
Minor
Line 109 and 152 – check the first word of the title
Need to improve the quality of Figures 3 and 4.
Author Response
Dear Reviewers:
We would like to thank you for taking time to review our manuscript. We have revised the manuscript according to your comments. We have sent the revised manuscript and highlighted the changes by red. The main corrections in the paper and the responses to the your comments are as following:
- Manuscript is very related to the current topic - preparation prebiotics based on degradation products of plant polysaccharides and is fully consistent with IJMS. The paper is well written, organized and adds new understanding to the xylo-oligosaccharides production. Abstract:Despite most of the research related to the impact of the resulting prebiotics on the microflora, the abstract provides an insufficient amount of first-time results.
Response:
Thanks for your comment. Compared with the colitis mouse (model group), XOS solution with higher X2-X3 fractions (S1 group) could significantly increase the number of Streptomyces in the intestinal microflora. While, the original XOS solution (S2 group) could significantly increase the number of Bacteroides in the intestinal microflora of colitis mouse. In addition, the abundances of Alcaligenes and Pasteurella in the intestinal microflora of in the intestinal microflora in the S1 and S2 groups were much lower than in the model group. We have added it to the abstract, see lines 25. Please check the addition.
- Introduction:Given the relevance of the ongoing research, there are not enough references in the introduction. An explanation is required why the authors are focused specifically on xylo-oligosaccharides, and not, say, for example, manno- derivatives? Why exactly X2-X3 needs to be explained in more detail. You can give an example of obtaining from raw materials as close as possible to bamboo.
Response:
Thanks for your comment. Xylooligosaccharides (XOS) are bioactive oligosaccharides formed through the β-1,4 glycosidic linkage of 2 to 10 xylose units. Xylobiose (X2), xylotriose (X3), xylotetracose (X4), xyloquinose (X5), and xylohexose (X6) constitute the primary constituents of XOS, in which X2 and X3 stand out as the principal bioactive components. XOS exhibits substantial research potential relative to other probiotics due to its cost-effectiveness, thermal and pH stability, sensory attributes, and multifaceted impacts on human health and livestock well-being. We have added it to introduction, see lines 38, 46, and 61. Please check the addition.
- Results and discussion:A discrepancy was found in the value of the component composition of raw materials Table 1 (first line) and Introduction (line 48).
Response:
Thanks for your comment. We have made changes in the introduction, see lines 61. Please check the changes.
- Materials and methodsIt is required to indicate in more detail what kind of enzyme preparation of xylanase was used, what is the activity? Industrial or laboratory? Were there any side activities?Were other (non xylo-) oligosaccharides identified?
Response:
Thanks for your comment. Jiangsu Kangwei Biotechnology Co., Ltd. provided the endo-xylanase (activity of 20 U/mL), which was derived from Trichoderma reesei. We have added it to 3.1. Material, see lines 432. Please check the changes.
- How was DNA isolated from microflora? What is the 16S rRNA gene fragment? On what equipment was sequencing carried out, according to what protocol?
Response:
Thanks for your comment. Specifically, the genomic DNA was extracted according to the instructions of Wizard® Genome DNA purification Kit (Promega). The purified genomic DNA was quantified by TBS-380 fluorimeter (Turner BioSystems Inc., Sunnyvale, CA). Genome sequencing combines PacBio RS II single molecule real-time sequencing (SMRT) and Illumina sequencing platform. Following bridge PCR amplification, double-ended sequencing (2×150bp) was done on the Illumina NovaSeq/Hiseq Xten (Illumina, USA) sequencing platform using the standard protocols in the analysis system. We have added it to Section 3.5, see lines 494-500. Please check the addition.
- Line 109 and 152 – check the first word of the title
Response:
Thanks for your comment. We have corrected it, see lines 124 and 166. Please check the changes.
- Need to improve the quality of Figures 3 and 4.
Response:
Thanks for Reviewer's comment. We have improved the quality of the Figure 3 and 4, see lines 379 and 422. Please check the changes.
Round 2
Reviewer 2 Report
The authors have made significant changes to the text of the manuscript and the quality of the drawings. Now I have no doubts about recommending this article for publication in IJMS.